# Methodology for Digital Synthesis of Deterministic and Random Jitter Generation on Rising or Falling Edges of Data Pattern

**Nan Ren** **, Zaiming Fu \*, Shengcun Lei, Hanglin Liu and Shulin Tian**

School of Automation Engineering, University of Electronic Science and Technology of China, Chengdu 611731, China; rennan@std.uestc.edu.cn (N.R.); lsc1995thoralex@163.com (S.L.); lnlhlzr@163.com (H.L.); shulin@uestc.edu.cn (S.T.)

\* Correspondence: fuzaimin@163.com; Tel.: +86-1808-018-0017

**Abstract:** Jitter is becoming an important factor in high-speed serial link and integrated circuits (ICs). Generating controllable jitter plays a crucial role in simulating the test environment of high-data links, evaluating the performance of IC, preventing jitter in high-speed serial link, and even testing the synchronous trigger circuit. In this paper, a digital synthesis for jitter generation and a logical combination method for selecting jitter on the rising edge or falling edge of a data pattern are presented. Precisely controllable jitter is generated by digital synthesis, including sinusoidal period jitter, rectangular period jitter, duty cycle distortion (DCD) jitter, and adjustable random jitter. Additionally, the validity and accuracy of the proposed method were demonstrated by hardware experiments, where the jitter frequency had an accuracy of ±30 ppm and the jitter amplitude had a step of 2 ps.

**Keywords:** high-speed serial link; integrated circuits (ICs); digital synthesis; logical combination; jitter generation

## 1. Introduction

Jitter is becoming an important factor in high-speed serial links and integrated circuits (ICs). New high-speed serial data standards are emerging, such as universal serial bus (USB) and the peripheral components interconnect express (PCIE). These serial standards are more susceptible to jitter and greatly cause bit error rate (BER) [1,2], and with them the requirement for effective compliance and characterization measurement [3,4]. As data rates for new generation IC continue increases, jitter injected module (JIM) is intended to generate controllable jitter, simulating the test environment of high-data links and evaluating the performance of IC [5–7]. In addition, the synchronous trigger circuit is also concerned with the jitter between the signals; however, it only considers the rising edge of signals. Jitter generation can perform jitter tolerance measurements by generating quantized controllable jitter and injecting different types of jitter into the incoming high-speed data stream, detect the BER of the code stream transmitted in the channel, and verify the ability of the receiver clock data recovery (CDR) while maintaining performance levels [8,9]. It can also be used to simulate the test environment of high-data links while evaluating the performance of the integrated circuit or system [5–7]; be used in CSC spread spectrum to reduce interference [10], improve the linearity of the CDR phase detector (PD) [11]; and measure the input of the phase interpolator for measuring the correlation jitter between data signal and CDR [12] and timing specifications for routers, gateways, or digital subscriber line access multiplexers (DSLAMs) [13].

In industry as well as in academia, the concept of jitter is varied (for a detailed introduction, please refer to [14,15]). The definition used in this paper is time interval error (TIE) jitter. Jitter refers to the

short-term variation of a digital signal from ideal positions at an important point in time. It can also refer to the deviation of an actual signal from the edge of an ideal signal, that is, the deviation of the time of the signal. Jitter can be characterized primarily by two typical jitters: random jitter (RJ) and deterministic jitter (DJ). DJ is bounded jitter, which can be divided into data-dependent jitter (DDJ), periodic jitter (PJ), and bounded uncorrelated jitter (BUJ). DDJ can be divided into duty cycle distortion (DCD) and intersymbol interference (ISI) [2,16].

References [17–19] describe analog modulation, using two signal generators to implement clock and modulation signals that act as jitter profiles first, and then a jittery clock is generated that is used as an external clock source to provide a clock for a pattern generator. Finally, the jittery data pattern is generated by data pattern and jittery clock. This method is complicated by multiple instruments, and the analog modulation introduces uncontrollable noise, which affects the accuracy of the generated jittered data signal, and the method cannot select the data edge to generate jitter. The method in reference [20] uses $\sum \Delta$ modulation, using a 1-bit ADC (analog-to-digital convert) or DTC (digital-to-time convert), and is first performed on the clock signal and the desired jitter profile and then by using the TMF (time mode filter) filters' quantization noise to generate jittery clock. Finally, the jittery clock signal and the data pattern are synthesized through a latch to obtain jittery data pattern. This method reduces the introduced noise via digital–analog combination, but the $\sum \Delta$ modulation still generates quantization noise, so a low-pass filter is required for filtering noise, which is complex and limits the frequency of the resulting jitter. This method also cannot select the edge of the data pattern to generate jitter. The present paper generates jitter by employing a digital synthesis method where the jitter amplitude depends on the delay and the jitter frequency depends on the frequency of variable clock. At the same time, the edge of the data pattern that produces jitter can be selected by XOR gate and ADD gate. This paper obtains the clock of the data pattern via clock and data recovery circuit first, and then realizes the digital synthesis of this clock controlled by the programmable delay line where the modulated signal is generated by the FPGA. The modulated clock and data pattern are connected to the D flip-flop to obtain a jittery data pattern.

The rest of this paper is organized as follows. In Section 2, the traditional method of jitter generation is compared with the method proposed in this paper. In Section 3, TIE, sinusoidal jitter, rectangular jitter, DCD, and random jitter are reviewed. In Section 4, the digital synthesis method and logical combination for jitter generation are described. In Section 5, measurement results show the validation of our proposed methods, and Section 6 presents the conclusions of the paper.

## 2. Traditional Jitter Modulation

This section describes the traditional method of jitter generation by analog modulation and compares it with the new method proposed in this paper.

As shown in Figure 1, the jittery clock is generated by two signal generators. The signal generator#1 serves as a modulation source to generate a modulated signal, and the signal generator#2 serves as a clock source to generate a jittery clock. It is used as the clock that generates the pattern to implement the jittery data pattern.

In order to facilitate the calculation, the sine wave is used for derivation in this paper. The final output signal is as follows:

$$s(t) = A_c \cos[w_c t + \theta(t)] \tag{1}$$

where $A_c$ is the amplitude of output signal, $w_c$ is the angular frequency of carrier, and $\theta(t)$ is modulation signal. According to the nature of the frequency modulation, the modulation signal is as follows:

$$\theta(t) = 2\pi D_f \frac{A_m}{2\pi f_m} \cos(2\pi f_m t - \frac{\pi}{2}) \tag{2}$$

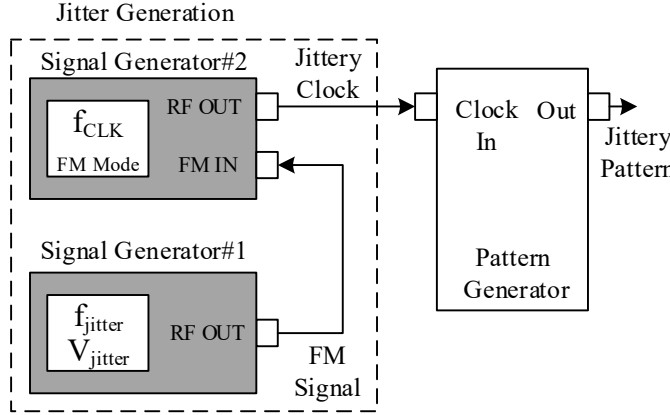

**Figure 1.** Analog modulation for jitter generation.

It is assumed that the modulated signal is a sinusoidal function, where $A_m$ is the amplitude of the modulation signal, $f_m$ is the frequency of the modulation signal, and $D_f$ is the frequency offset constant. Then the peak-to-peak value of phase offset is

$$\theta(t)_{p-p} = 2 * 2\pi D_f \frac{A_m}{2\pi f_m} = \frac{2D_f A_m}{f_m} \tag{3}$$

The magnitude of the jitter obtained according to the definition of jitter is (in UI)

$$A_{jitter} = \frac{\theta(t)_{p-p}}{2\pi} = \frac{D_f A_m}{\pi f_m}[UI_{p-p}] \tag{4}$$

According to Equation (11), it can be seen that the factors affecting the amplitude of the jitter generated by the analog modulation are $D_f$, $f_m$, and $A_m$; this means that adjustable jitter amplitude is limited by jitter frequency. When the jitter frequency increases, the jitter amplitude decreases. The jitter amplitude is at least 0.1 Uipp (at a jitter frequency of 20 M) in [19], and jitter amplitude is at least 0.01 Uipp (at a jitter frequency of 2 M) in [21], but, in this design example, the jitter amplitude can be adjusted freely and its range of 10 ps–10 ns, a step of 2 ps or 10 ps, is independent of jitter frequency and data rate, as shown in Table 1. The frequency resolution of jitter using analog modulation is ±50 ppm; however, the jitter frequency resolution in this design example is ±30 ppm in the actual test, as shown in Table 1.

**Table 1.** Comparing jitter generation.

|  | [21] | [19] | This Paper |
|---|---|---|---|
| Jitter frequency | 1 Hz–100 MHz | 1 kHz–20 MHz | 1 Hz–20 MHz |
| Jitter frequency Resolution | 1 Hz in 1 Hz to 1 MHz; 1 kHz in 1 MHz to 100 MHz | / | 1 Hz in 1 Hz to 1 MHz; 1 kHz in 1 MHz to 20 MHz |
| Jitter frequency accuracy | ±50 ppm | / | ±30 ppm |
| Minimum jitter amplitude | 0.6 UI (100 M jitter frequency) | 0.2 UI (20 M jitter frequency) | 10 ps (Not related to jitter frequency and data rate) |
| Minimum jitter amplitude resolution | 0.01 UI (100 M jitter frequency) | / | 10 ps (digital control) 2 ps (analog control) |

## 3. Jitter Modeling

In this section, we first give the TIE jitter model, and then give the specific model of PJ and DDJ according to TIE. The periodic jitter model of rectangular, sine waves, DCD, and ISI, in time domain, is given.

### 3.1. Time Interval Error Jitter

In a high-speed serial data system, TIE jitter refers to the phase difference between the edge of the data signal and the edge of the clock signal, as shown in Figure 2 [22]. When the phase deviation is significant, setup time, insufficient hold time, and bit error may occur.

$$TIE_k = t_{Dk} - t_{Ck} \tag{5}$$

where $t_{ck}$ is the time of each rising edge of the clock and $t_{Dk}$ is the time of the data pattern edge position corresponding $t_{ck}$.

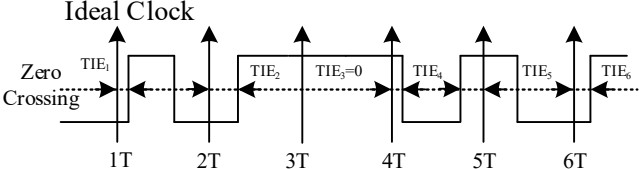

**Figure 2.** Digital synthesis.

### 3.2. Jitter Components

In this paper, the random jitter and deterministic jitter are generated by digital synthesis. As a result, the models of jitter components need to be created. They include models for sinusoidal jitter, rectangular jitter, duty cycle distortion jitter, and random jitter.

#### 3.2.1. Period Jitter

The period jitter is the deviation of the cycle time from the ideal period [23], and may also refer to the jitter signal [24] that recurs over a certain period or frequency. Periodic jitter, sometimes referred to as sinusoidal jitter, is uncorrelated with the data signal and it is caused by adjacent circuits such as power supply noise, on-chip oscillators, and data buses. The most common time domain model for periodic jitter is sinusoidal jitter, which is often used for jitter tolerance testing, but rectangular jitter is also important for bit error rate.

Here, a sinusoidal jitter is given by the following equation [25]:

$$\begin{aligned}
\triangle t_{\text{SIN}}[n] \quad &= A\sin(2\pi f_0(t - nT) + \Phi) \\
&= a\sin\left(\frac{2\pi f_0 n}{f_s}\right) + b\cos\left(\frac{2\pi f_0 n}{f_s}\right)
\end{aligned} \tag{6}$$

where $\triangle t_{\text{SIN}}[n]$ is sinusoidal jitter amount at sampling time $nT$, $f_0$ represents the frequency of sinusoidal jitter, and $A$ is the amplitude of sinusoidal jitter.

The mathematical model of the rectangular period jitter is given by the following equation:

$$\triangle t_{\text{rect}}[n] = A_{\text{rect}} \times \text{sgn}[\sin(2\pi f_j nT)] \tag{7}$$

where $nT$ is the sampling time of the rectangular jitter, $A_{\text{rect}}$ is the amplitude of the rectangular jitter, $f_j$ is the frequency of the rectangular jitter, and sgn[•] is the rectangular wave function

$$\text{sgn}[T] = \begin{cases} 1 & T \geq 0 \\ -1 & T < 0 \end{cases} \tag{8}$$

### 3.2.2. Duty Cycle Distortion Jitter (DCD)

DCD is an important deterministic jitter. It is caused by non-idealities such as asymmetric rising and falling edges of the path, and it is half of data rate, which can be modeled as [25]

$$
\begin{aligned}
\Delta t_{\text{DCD}}[n] &= J_{\text{DCD}} \times \cos(n\pi) \\
&= [-J_{\text{DCD}}, J_{\text{DCD}}, -J_{\text{DCD}}, J_{\text{DCD}}, \dots]
\end{aligned}
\tag{9}
$$

where $\Delta t_{\text{DCD}}[n]$ is the DCD of the sampling time $nT$, and $J_{\text{DCD}}$ is the amplitude of the DCD.

### 3.2.3. Random Jitter (RJ)

RJ is caused by thermal noise, shot noise, and other high-order noise. It can be created by Gaussian white noise. And the statistical probability density function (PDF) for RJ is given by [26]

$$
f_{\text{RJ}}(\Delta t) = \frac{1}{\sqrt{2\pi}\sigma} \exp -\frac{(\Delta t - \mu)^2}{2\sigma^2}
\tag{10}
$$

where $f_{\text{RJ}}(\Delta t)$ is RJ amplitude of data bit $n$ at sample time $nT$, $\mu$ is the mean of RJ, and $\sigma$ is the standard deviation of RJ. In this method, the $\sigma$ can be adjusted.

## 4. Jitter Generation Method

This section details the methods of jitter generation, including digital synthesis and logical combination method for high-precision jitter generation.

Figure 3 shows that the digital synthesis method of generating controllable jitter can be divided into clock and data recovery circuit (CDR), jitter generation, delay compensation, jitter synthesis, and logical combination. The method is described in detail referring to the jitter generation block diagram.

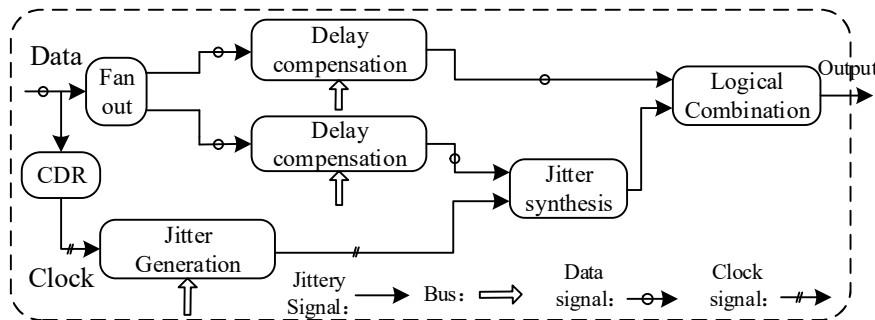

**Figure 3.** Digital synthesis method of generating controllable jitter.

The core idea of this paper has two points. The first point is to use digital synthesis to generate jitter. It is necessary to generate delay via a field-programmable gate array (FPGA) and programmable delay line for digital synthesis, where FPGA controls jitter frequency and programmable delay line controls jitter amplitude. The second point is to use logical combination to select the signal edge of generating jitter. The jitter of the falling edge is directly generated by the AND gate, and the jitter of the rising edge is excited by the XOR gate and then through AND gate. The data pattern is obtained by fanning out from the input data pattern when selecting signal edges.

The jitter generation circuit generates jittery clock by using a field-programmable gate array (FPGA) and programmable delay line, which control frequency and amplitude of jitter separately. This circuit is mainly divided into three parts, as shown in Figure 4: data processing, digital synthesis, and programmable delay. Data processing is the analysis of the data sent by the host computer; digital synthesis includes variable clock, address generation, ROM waveform, and amplitude processing; and the delay line implements a quantitative delay of the clock signal for jitter generation. The frequency

of the jitter generation is determined by the clock generating the address and the number of data in the ROM, and the amplitude of the jitter generation is determined by the value of the delay of the delay line.

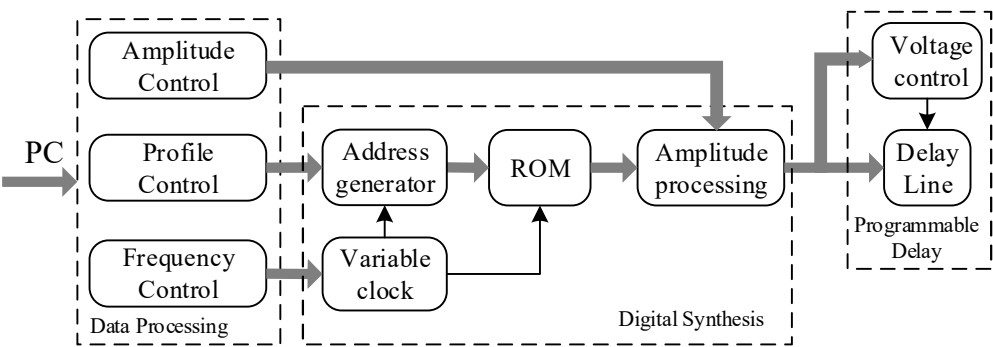

**Figure 4.** Jitter generation circuit.

The data processing is to process the address and command sent by the host computer. The different addresses correspond to the amplitude, frequency, and profile control, respectively, and their commands have different control modes. When the frequency control is performed, a control word is sent to control variable clock to change the clock output. When the profile control is performed, the value of the address generator is changed and the different profile of jitter required to be stored in the ROM is read, and when the amplitude control is performed, the amplitude of the data extracted from the ROM is processed by changing the magnitude of the delay.

Digital synthesis consists of a variable clock generated by phase-locked loop, an address generator, a ROM look-up table, and amplitude processing. The clock generated by the variable clock is used to control the address generator to generate an address for reading the jitter model in the ROM lookup table and send the jitter model to the amplitude processing. The mathematical jitter model in the ROM lookup table can be modified to any desired jitter model, and the jitter model in this design is a mathematical model of pre-set period jitter. The frequency of the variable clock determines the frequency of the jitter generation. The phase-locked loop used in this design produces a variable clock with a resolution of 1 millihertz (mHz). Therefore, the resolution of the frequency of jitter generation is 1 millihertz (mHz).

Programmable delay is used to achieve the clock jitter generation. The programmable delay line used in this design is divided into analog control and digital control. The digital control port of the programmable delay chip implements a minimum delay resolution of 10 ps, the minimum resolution of the analog control is 2 ps, and the analog input of the programmable delay line is controlled and calibrated by a high-precision digital–analog convertor with very fine analog output steps. Therefore, the minimum resolution of the jitter generation is 2 ps.

There is a design example of jitter generation, which generates a rectangular period jitter with a frequency of 10 Mega-Hertz (MHz) and an amplitude of 20 picoseconds. First, we need to control the phase-locked loop through FPGA to generate a 200 MHz sampling clock, and then, using this sampling clock, to generate addresses and read the jitter data in ROM by generated addresses, where the data model of rectangular period jitter in ROM is generated by formulas (7) and (8). Finally, the data are calculated by the amplitude factor generated by the amplitude control in Figure 3 to obtain the final delay control. The timing diagram of this design example is shown in Figure 5. We assume that ROM_data has 20 data in Figure 5, the ROM_data is sent by a 200 MHz sampling clock, and the jitter frequency obtained is 10MHz. Because the amplitude of rectangular period jitter in ROM_data is 1 in Figure 5, and the digital step of the programmable delay controlled by Delay_ctl is 10 picoseconds, we need that the amplitude factor is 2 if the amplitude of the jitter generated is 20 picoseconds. Delay_ctl is the output of the FPGA and the input of the programmable delay.

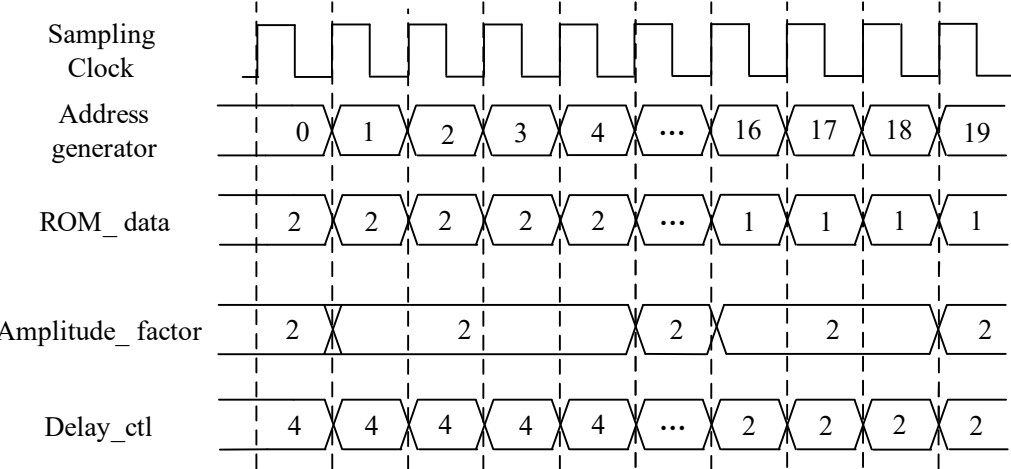

**Figure 5.** Timing diagram of jitter generation.

Different from the deterministic jitter generation, the generation of a random jitter data stream is special. The random jitter data stream is generated by MATLAB, and the Gaussian white noise function *randn*, the constant function, and the signal-to-noise ratio (SNR) of the signal are required. It can also be generated by other algorithms, such as the Box–Muller algorithm, and the central limit theorem (CLT) method. [27,28]

The definition of SNR is as follows:

$$SNR = 10\log\frac{P_s}{P_n} \tag{11}$$

where $P_s$ is the power of the signal and $P_n$ is the power of the noise. According to the definition of Gaussian white noise, $P_n$ is also the variance of Gaussian white noise [29]. We can generate the original signal via the constant function; Gaussian white noise via the randn function; calculate $P_s$, $P_n$, and SNR; and, finally, get the relationship between the variance and other known quantities, as shown in Equation (12). The specific process of generating a random jitter data stream is shown in the Figure 6.

$$\sigma^2 = P_n = \frac{P_s}{\frac{1/10SNR}{10}} = \frac{P_s}{SNR} \tag{12}$$

Assuming that the Gaussian noise signal sent to ROM is $y_i(i = 0, 1, 2, \ldots, 1023)$, the mean is $\mu_0$, and the root mean square is $\sigma_0$; then, the final output $\sigma$ through jitter generation circuit is

$$\sigma = \sqrt{\frac{(Ny_i - N\mu_0)^2}{1024}} = N\sigma_0 \tag{13}$$

where $N$ is the amplitude factor generated by the amplitude processing in Figure 4. When the random jitter is generated, the random jitter data stream in ROM is read by the sampling clock that is generated by the PLL, and it is sent to the amplitude processing and multiplied by the amplitude factor. Finally, the result is used as a control word of the programmable delay line to generate a controllable random jitter.

The delay compensation is used to compensate the phase deviation to ensure that the phase of the data pattern and the jittery clock are the same when the jitter is synthesized [4]. The compensation can use the FPGA count delay or the delay chip. Because the FPGA count delay is limited by the counter operating frequency, the resolution of the count delay is generally of nanoscale order. At the same time, considering the phase compensation calibration, the programmable delay chip is used in this design for delay phase compensation and improves the resolution of delay.

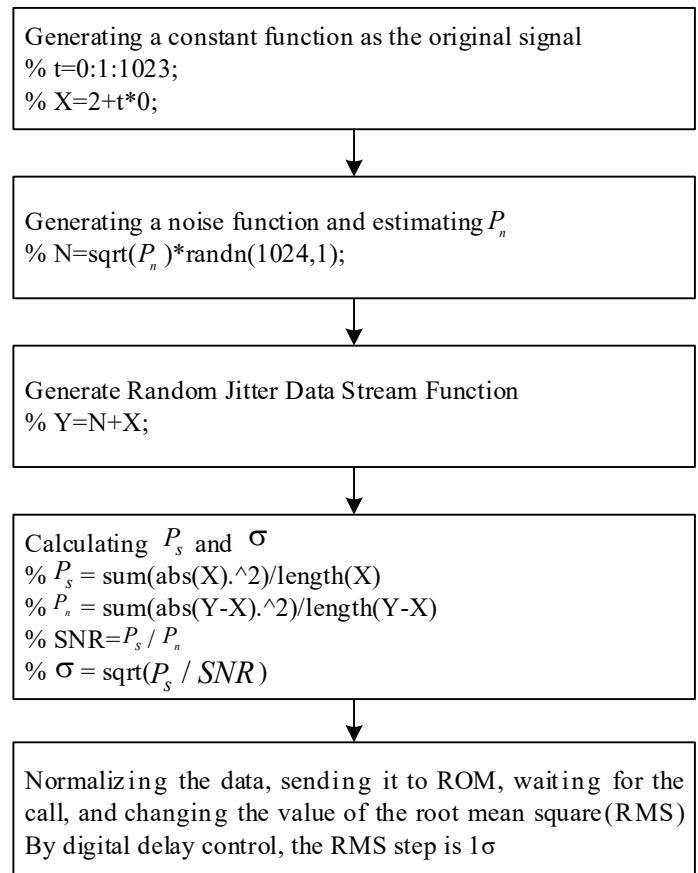

**Figure 6.** The specific process of generating a random jitter data stream.

The jitter synthesis in this design example is implemented by high-speed D flip-flop, and its principle is a relative delay by using the clock side and D side of the D flip-flop, as shown in Figure 7, and the data pattern and the jittery clock are, respectively, connected to the D side and the clock side. The D flip-flop output side Q is pulled down to a low level or a high level depending on the data pattern level when the rising edge of jittery clock comes. For example, if the data pattern is at a high level when the rising edge of jittery clock comes, the Q side will be pulled down to a low level till next rising edge of jittery clock. In this way, jittery data pattern is synthesized because of jittery clock.

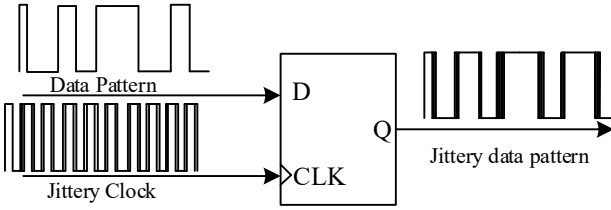

**Figure 7.** Schematic of the principle of the jitter synthesis.

The logical combination consists of a high-speed AND gate, a XOR gate, and three selectors, as shown in Figure 8, which are used to select the edge of jitter generation. It is necessary to decompose the input data pattern of the system into two channels that are separately delayed to generate jittery data pattern and select edge of jitter generation. The data pattern used to generate jittery data pattern and the data pattern used to select the edge of jitter generation are obtained by fanning out from the input data pattern, because the final rising or falling jittery data pattern is synthesized using logic

operation of the jittery data pattern and the data pattern used to select the edge of jitter generation, as shown in Figure 8.

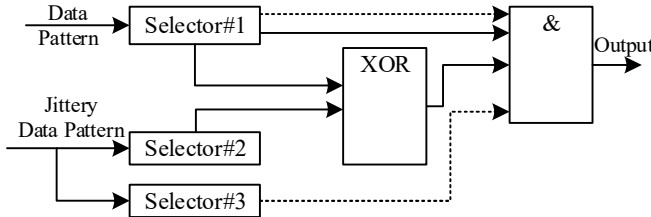

**Figure 8.** Method of jitter generation on rising or falling edge.

When the jitter on the rising edge of data pattern is to be generated, as shown by the solid line in Figure 8, selector#1 fans out two data patterns, which are used as input of the XOR gate and ADD gate, respectively, and selector#2 fans out jittery data pattern from the output of the D flip-flop as the other input of the XOR gate. The output of the XOR gate is the other input of the AND gate for generating jitter on the rising edge of data pattern. The timing diagram of jitter generation on the rising edge is shown in Figure 9, where the D Flip-flop represents the output of selector#2 and the pattern represents the two outputs of selctor#1.

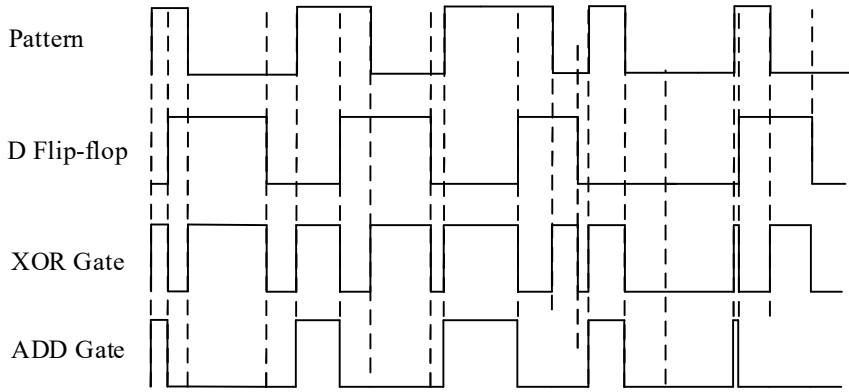

**Figure 9.** Time diagram of jitter generation on rising edge.

When the jitter on the falling edge of data pattern is to be generated as shown by the dotted line in Figure 8, selector#1 fans out data pattern, which is used as input of the ADD gate directly, and selector#3 fans out jittery data pattern as the other input of the ADD gate for generating jitter on the falling edge of data pattern. The timing diagram is shown in Figure 10, where the D Flip-flop represents the output of selector#3 and the pattern represents the output of selector#1.

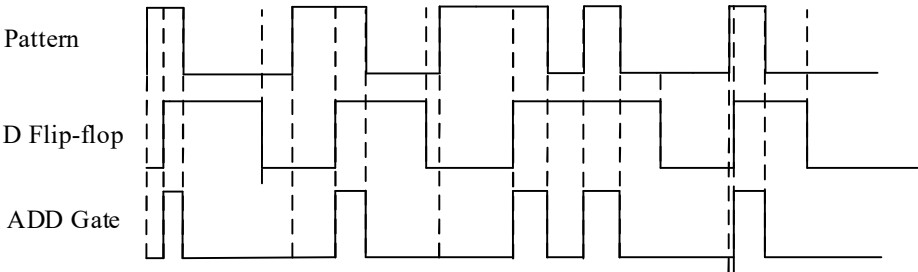

**Figure 10.** Time diagram of jitter generation on falling edge.

## 5. Measurement Results

### 5.1. System of Experiment

Figure 11 shows the experimental physical test diagram. The experimental test equipment consists of an oscilloscope, which has a bandwidth of 13G, a sampling rate of up to 40 Gb/s, a memory depth of up to 768 Mm, and pulse pattern generator Agilent 81130A, which is used to generate data patterns and clocks, as well as jitter generation modules. The oscilloscope has a full suite of tools for serial data analysis, debugging, verification, and compliance testing. SDA III eye diagram and jitter analysis software can implement serial-data jitter analysis, including jitter separation using the Dual Dirac model; analysis of jitter using the histogram and TIE track; and measurement DCD, RJ, and PJ.

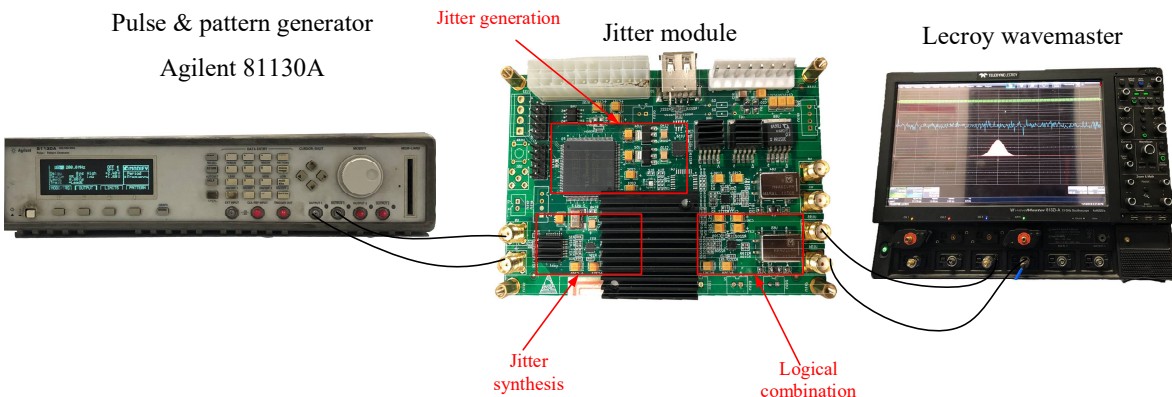

**Figure 11.** Installed system of the experiment.

### 5.2. Validation of Jitter Generation

The jitter generation in this design is mainly based on the TIE jitter in the time domain model. Therefore, the results focus on its TIE track and histogram with an oscilloscope.

The bathtub curve, eye diagram, TIE track, and histogram of sinusoidal jitter are shown in Figure 12, where test signal frequency is 200 MHz, the jitter frequency is 10.27 MHz, the period jitter is 23.972 ps, and the random jitter is 1.737 ps. Generated sinusoidal jitter can simulate jitter caused by power supply noise, on-chip oscillators, etc. It can also be used for jitter tolerance testing. The histogram and TIE track of rectangular jitter are shown in Figure 13, where test signal frequency is 400 MHz, the jitter frequency is 1.632 kHz, the period jitter is 1.0339 ns, and the random jitter is 2.3 ps. The histogram and TIE track of DCD jitter is shown in Figure 14, where test signal frequency is 200 MHz, the jitter frequency is 194.753 MHz, the DCD jitter is 812.021 ps, and the random jitter is 2.1 ps. DCD jitter is only for clock patterns of repeating 0101 bits, and its frequency is correlated with the data rate. The TIE track of DCD generated by Equation (9) is a triangular wave composed of two points as shown in Figure 12, and the histogram of DCD can be seen as the Dual-Dirac delta function. Generated DCD jitter can simulate DCD during signal transmission. The histogram and TIE track of Random jitter are shown in Figure 15, where test signal frequency is 200 MHz, and the root mean square (RMS) of RJ is 436 ps. The adjustable range of RJ is from 14 ps to 8.33 ns, and generated RJ can simulate noise environment during signal transmission.

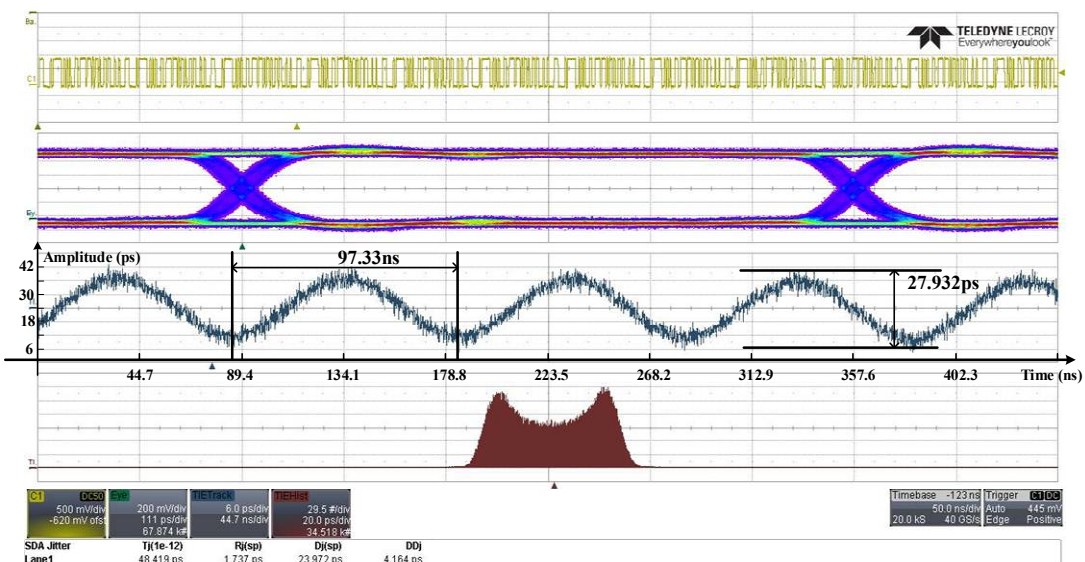

**Figure 12.** Eye diagram, time interval error (TIE) track and histogram of sinusoidal jitter.

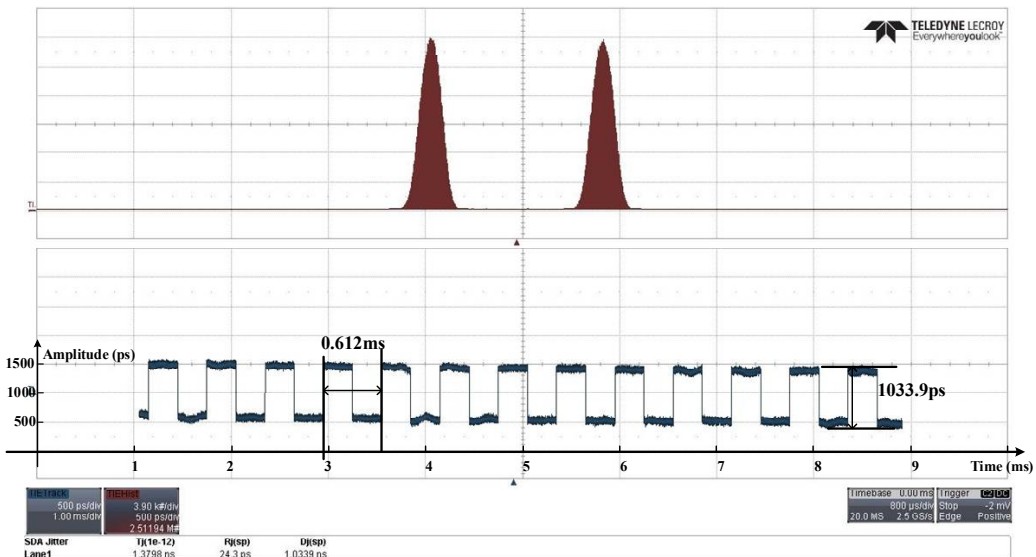

**Figure 13.** Histogram and TIE track of rectangular jitter.

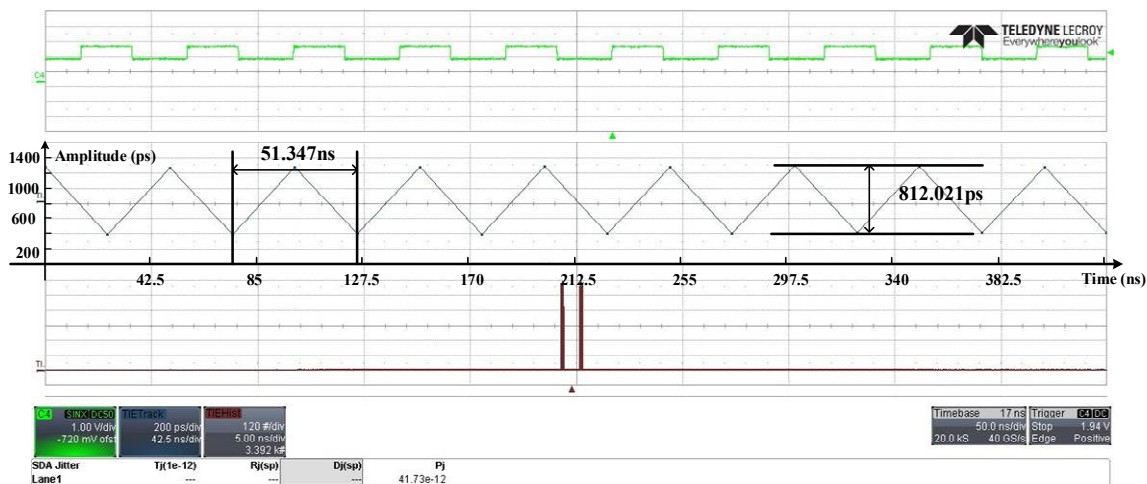

**Figure 14.** TIE track and histogram of duty cycle distortion (DCD) jitter.

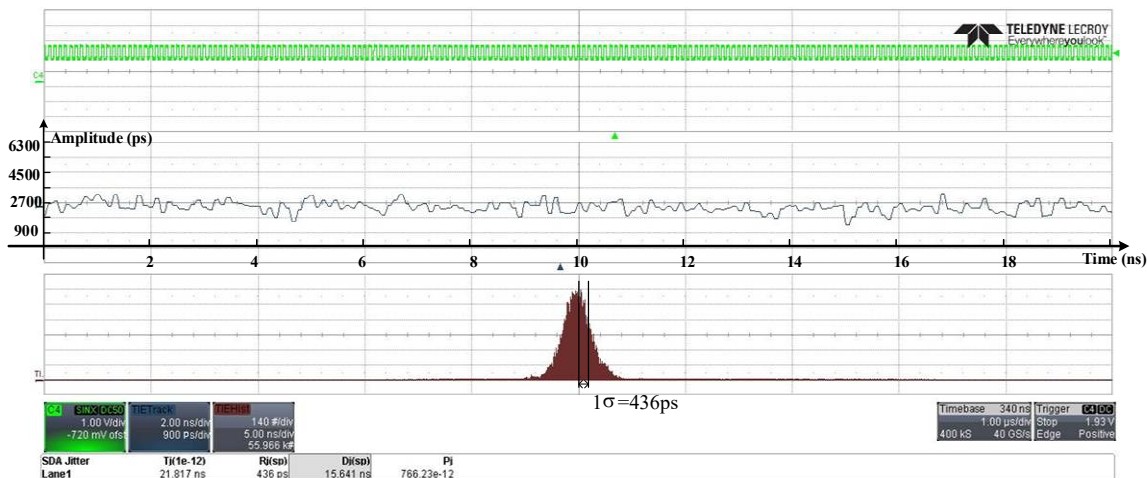

**Figure 15.** TIE track and histogram of random jitter.

Jitter generation on the rising of data pattern is shown in Figure 15; the green signal is original signal and the pink signal is jittery signal on the rising of data pattern. According to the definition of TIE jitter, the difference between two solid lines in Figure 16 is the jitter, and the dotted line in Figure 15 indicates that the falling edges of two signals are consistent. The phase difference between the rising edges of the green and pink signals can be used to test or implement a synchronous trigger circuit. Jitter amplitude encompasses a range of 12ps–(cycle*0.1) ps, a step of 2 ps.

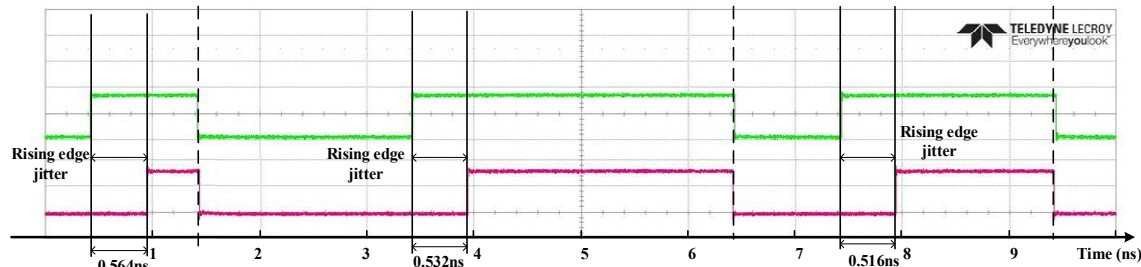

**Figure 16.** Jitter generation on rising edge, the green signal is original signal and the pink signal is jittery signal and the difference between two solid lines is TIE jitter

## 6. Conclusions

This paper proposes a full digital synthesis method that achieves controllable jitter generation and a logical combination method that selects jitter on the rising edge or falling edge of a data pattern. This method realizes deterministic jitter and random jitter generation, including sinusoidal period jitter, rectangular period jitter, DCD jitter, and adjustable random jitter. It can also generate a jittery rising signal for the synchronous trigger circuit. The proposed digital synthesis jitter method can be equivalent to converting controllable digital delay to jitter, and the jitter resolution is greatly improved by using a programmable delay line for precise control of the jitter generation. In the experimental test, it can be seen that the amplitude of jitter accuracy can reach 2 ps and the jitter frequency resolution is ±30 ppm. The method can be applied to jitter tolerance test, jitter simulation, and synchronous trigger circuit.

**Author Contributions:** Conceptualization, N.R. and Z.F.; methodology, N.R.; software, H.L.; validation, S.L., N.R. and Z.Z.; formal analysis, N.R.; investigation, S.T.; resources, Z.F.; data curation, S.L.; writing—original draft preparation, N.R.; writing—review and editing, Z.F.; visualization, N.R.; supervision, H.L.; project administration, H.L.; funding acquisition, Z.F.

**Funding:** This research was funded by National Natural Science Foundation of China under Grant No. 61871089 and Grant 61671114.

**Acknowledgments:** Thanks my family. Thanks my girlfriend. Thanks to Liu Jun for his help in mathematics derivation. Thank you to all the people who support us.

**Conflicts of Interest:** The authors declare no conflicts of interest.

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
