# Peer review of "Methodology for Digital Synthesis of Deterministic and Random Jitter Generation on Rising or Falling Edges of Data Pattern"

_electronics, doi:10.3390/electronics8121510_

Round 1

Reviewer 1 Report

Generally well written and easy to follow. Maybe add more references about applications and backgrounds. 

Reviewer 2 Report

This manuscript presented the methodology to generate jitter patterns. After reviewing this manuscript, I found that there are some problems as follows:

1. Why didn't you present the 'traditional jitter modulation' (part 4) after the introduction part, or as a background part?

2. I didn't find any application of the 'jitter modeling' part to the rest of the manuscript.

3. I cannot find any relationship between Fig. 2, Fig. 3 and Fig. 9.

4. At lines 143 and 144, 1mHz is one millihertz or Mega-Hertz?

5. In Fig. 9, the model of the pulse generator is not correct.

6. What is the product model of the jitter module?

7. Which function is implemented in the FPGA?

8. How to implement the amplitude processing and programmable delay functions on an FPGA in Fig. 2? Please show the timing diagram of these functions to point out their relationship.

9. Why did you have two output ports connected to the oscilloscope?

10. At line 236, the test signal frequency is 200 MHz, but it is not true in Fig. 10.

11. Please show the time axes in Figs. 10 to 14.

12. How to generate the random jitter in Fig. 13 since I don't find any circuit to implement it.

13. I cannot find any jitter in Fig. 14.

14. What is the purpose of the first window (yellow) in Fig. 10?

15. What is the advantage of this work compared to the state-of-the-art, for example, Ref. [20] and Ref. [13]?

Round 2

Reviewer 2 Report

The authors solved all the problems. The revised manuscript looks better.

Author Response

Thank you for your comment. Now I add the actual method implemented to obtain an AWGN stream and describe the method of random jitter generation.